# Transcription factor interactions explain the context-dependent activity of CRX binding sites

Kaiser J. Loell[1,2], Ryan Z. Friedman [1,2], Connie A. Myers[3], Joseph C. Corbo[3], Barak A. Cohen[1,2], Michael A. White [1,2]*

1 Department of Genetics, Washington University School of Medicine in St. Louis, St. Louis, Missouri, United States of America, 2 The Edison Family Center for Genome Sciences & Systems Biology, Washington University School of Medicine in St. Louis, St. Louis, Missouri, United States of America, 3 Department of Pathology and Immunology, Washington University School of Medicine in St. Louis, St. Louis, Missouri, United States of America

* mawhite@wustl.edu

## Abstract

The effects of transcription factor binding sites (TFBSs) on the activity of a *cis*-regulatory element (CRE) depend on the local sequence context. In rod photoreceptors, binding sites for the transcription factor (TF) Cone-rod homeobox (CRX) occur in both enhancers and silencers, but the sequence context that determines whether CRX binding sites contribute to activation or repression of transcription is not understood. To investigate the context-dependent activity of CRX sites, we fit neural network-based models to the activities of synthetic CREs composed of photoreceptor TFBSs. The models revealed that CRX binding sites consistently make positive, independent contributions to CRE activity, while negative homotypic interactions between sites cause CREs composed of multiple CRX sites to function as silencers. The effects of negative homotypic interactions can be overcome by the presence of other TFBSs that either interact cooperatively with CRX sites or make independent positive contributions to activity. The context-dependent activity of CRX sites is thus determined by the balance between positive heterotypic interactions, independent contributions of TFBSs, and negative homotypic interactions. Our findings explain observed patterns of activity among genomic CRX-bound enhancers and silencers, and suggest that enhancers may require diverse TFBSs to overcome negative homotypic interactions between TFBSs.

## Author summary

Transcription factors control gene expression in different cell types by binding to sites in regulatory DNA. The same transcription factor when bound at different DNA sites will have different effects on gene expression, but how a single factor can produce divergent effects is unclear. The photoreceptor transcription factor CRX activates expression from regulatory DNA that harbors few copies of a CRX binding site, while it represses expression when many binding site copies are present. We modeled how the number and arrangement of binding sites for CRX and other factors affect gene expression, using data

**Data Availability Statement:** All processed data files and code are available at https://github.com/mawsterlab/synthetic_promoter_mavenn. FASTQ files for the MPRA experiments with the CDNRM

library are deposited with the NCBI Gene Expression Omnibus (GEO) under accession number GSE225867.

**Funding:** This work was supported by National Institutes of Health grants R01 GM121755 to M.A. W.; R01 GM092910 to B.A.C.; EY030075, HL149961, and MH122451, to J.C.C.; and F31HG011431 to R.Z.F. The funder played no role in the study design, data collection and analysis, decision to publish, or preparation of the manuscript.

**Competing interests:** I have read the journal's policy and the authors of this manuscript have the following competing interests: BAC is on the scientific advisory board of Patch Biosciences. Neither any reagent nor any funding from this organizations was used in this study. Other co-authors have no competing interests to declare.

from libraries of synthetic regulatory DNA elements. The model shows that individual transcription factor binding sites increase expression on their own, but interactions between multiple copies of the same site decrease expression. Our results generalize across transcription factors and tissues, suggesting that this is a general principle that might help explain differing patterns of expression across tissues. The model explains how interactions between binding sites allow a single transcription factor to have contrasting effects on gene expression in the same cell type.

## Introduction

A typical mammalian transcription factor (TF) binds hundreds or thousands of *cis*-regulatory elements (CREs) in the genome [1–3]. CREs that are bound by the same TF vary widely in their activity, and can include strong enhancers, transcriptional silencers, or sequences with weak or no *cis*-regulatory activity [3–20]. Such dramatic functional differences among CREs with similar TF binding sites (TFBSs) show that local sequence context modulates the contribution of a TFBS to *cis*-regulatory activity, yet how this occurs is not well understood. When accounting for context dependence, proposed models of *cis*-regulatory grammar vary in their emphasis on the importance of interactions between TFs, and they suggest different degrees of flexibility in the possible functional arrangements of TFBSs [21–24]. The enhanceosome model proposes that strict geometrical constraints determine whether CRE-bound TFs can activate transcription. This implies that context-dependent effects of TFBSs are strongly influenced by highly specific interactions between them [21–23,25]. The contrasting billboard model proposes that active CREs are defined by the presence of a sufficient number of bound TFs, with no strong constraints governing their arrangement [24]. The billboard model implies that the context of a TFBS is determined primarily by additive effects of the surrounding TFBSs, with few specific interactions between sites. Taking an intermediate position between enhanceosome and billboard models, the TF collective model proposes that cooperative interactions between TFs are important, but that these interactions do not depend on a specific motif grammar [26]. Other models of *cis*-regulatory grammar propose that individual TFBSs are weak on their own and depend on strong cooperative interactions [27], that particular TFBSs recruit specific, required transcriptional cofactors [11], or that the balance among sites for transcriptional activators and repressors determines whether a CRE is an enhancer or silencer [28–30]. An effort to harmonize the range of motif flexibility observed in natural enhancers is the dependency grammar model, which recognizes that the presence or absence of constraint on motif identity, affinity, and arrangement likely depends on an interplay between these features [21]. Under this model, different enhancers will vary in how strongly their activity depends on strict rules of motif composition. However, how the proposed features of *cis*-regulatory grammar work together to define the local context of a CRE remains unclear. As a result, accurately predicting the activity of CREs or the effects of genetic variants in TFBSs is an unsolved problem.

Local sequence context has strong effects on the function of binding sites for the retinal TF Cone-rod homeobox (CRX) [31–33]. CRX is a paired-type K50 homeodomain TF and a critical regulator of transcription in multiple retinal cell types, where it contributes to both activation and repression of cell type-specific genes [8,31,33–42]. Using massively parallel reporter assays (MPRAs) conducted in mouse retinal explants, we previously found that genomic CRX-bound sequences include strong enhancers and silencers [13,18,43,44]. The activities of these CREs, whether activating or repressing, depend on both CRX binding sites and CRX protein,

which demonstrates that the effects of CRX sites are modulated by context [13,43]. We showed that CRX binding sites can have opposite effects in different contexts: mutating CRX binding sites in genomic enhancers reduces MPRA activity, while mutating CRX binding sites in genomic silencers increases it [13,18,43]. While the critical role of local sequence is clear, the ways in which context determines whether a CRX-bound region functions as an enhancer or a silencer is not well understood. We previously identified features that partially distinguish CRX-bound enhancers from silencers. CRX cooperatively interacts with the rod photoreceptor-specific leucine zipper TF NRL at some rod gene promoters, and we showed that synthetic CREs with sites for CRX and NRL were often strongly activating [13,31,37,39,40,45–47]. However, an NRL site is not present at most CRX-bound enhancers and is thus not required for strong activity [18]. Compared to enhancers, genomic CRX-bound silencers tend to contain more copies of the CRX motif [13,18,43], while CRX-bound enhancers are enriched in sites for other TFs relative to silencers [18]. To account for these observations, we hypothesized that interactions between CRX and other co-bound TFs determine whether a sequence functions as an enhancer or silencer. We sought to capture those interactions in a quantitative model trained on data from synthetic CREs with defined binding sites for CRX and other photoreceptor TFs.

A key advantage of synthetic CREs is that their binding site composition can be systematically varied to generate informative training data for interpretable models of *cis*-regulatory grammars. We previously trained statistical thermodynamic models on data from reporter gene libraries of synthetic CREs with pre-defined TFBSs, in order to learn how interactions between TFBSs contribute to *cis*-regulatory grammars in a variety of cellular systems [16,48–51]. A major advantage of thermodynamic models is their interpretability, because their parameters represent biophysical TF-TF and TF-DNA interactions [52–54]. However, such models can be difficult to train successfully and often require computationally intensive custom fitting pipelines [53,55,56]. A recent general-purpose modeling framework, called MAVE-NN, overcomes these challenges using a neural-network based approach to fit interpretable genotype-phenotype maps to data from massively parallel functional assays [57]. An important difference between MAVE-NN and recent deep learning models such as Deep-STARR and others [58–61] is that MAVE-NN explicitly models the relationship between sequence and activity separately from features of the experimental measurement, such as saturation, detection limits, and noise, rather than attempting to model all features of an MPRA dataset in a monolithic architecture trained end-to-end. This enables MAVE-NN models to learn interpretable parameters that correspond straightforwardly to additive contributions and interactions between sequence features such as TFBSs. MAVE-NN is therefore well-suited to model datasets from synthetic CREs composed of pre-defined TFBSs

We used MAVE-NN to train models on data from MPRA libraries of photoreceptor-specific, synthetic CREs assayed in live retinal explants. We find that the effects of CRX sites are explained by a model that includes positive, additive contributions of individual TFBSs, negative homotypic interactions between sites for the same TFs, and positive heterotypic interactions between sites for different TFs. The model explains the observations that CRX sites produce context-dependent activation and repression, and that the addition of an NRL site converts silencers to enhancers. The model also accounts for our finding that CRX-bound enhancers have sites for a diverse set of TFs, while CRX-bound silencers lack this TFBS diversity. More generally, our results suggest that context-dependent activity of binding sites for transcriptional activators can be explained by the balance between the negative effects of interactions between sites for the same TF, the positive effects of individual TFBSs, and heterotypic cooperativity between sites.

## Results

### Positive heterotypic and negative homotypic interactions explain the effects of CRX and NRL sites on CRE activity in photoreceptors

We previously reported that both genomic and synthetic CREs with many binding sites for CRX tend to act as silencers, while CREs with fewer CRX sites tend to act as enhancers in a retinal explant MPRA [13,18]. Our prior results from a reporter library of 1,299 synthetic CREs showed that sequences composed of only CRX and NRL binding sites exhibit activity that ranges from strong activation to repression [13]. These CREs were tested by MPRA in mouse retinal explants, which preserve all retinal cell types and cell type-specific TFs that comprise the native context in which CRX is active. Sequences included up to four sites in either the forward or reverse orientation. TFBSs included high, medium, and low affinity versions of CRX sites and the consensus site for NRL (Fig 1A). CREs were cloned upstream of either the murine *Rho* or *Hsp68* basal promoter. The library included all 584 possible combinations of one, two, and three TFBSs, and 715 sequences randomly sampled from all possible combinations of four sites. Sequential addition of CRX sites upstream of a basal promoter led first to increased activation and then to repression below basal levels when three or four CRX sites were present (Fig 1A and 1B). Repressive CREs with four CRX sites could be converted to strongly activating sequences by replacing one CRX site with a site for NRL. Synthetic sequences composed of multiple CRX sites and one NRL site were more active than equal length CREs composed of only CRX sites (Fig 1B) or only NRL sites (S1A Fig). We found that genomic CRX-bound sequences followed a similar pattern [13]. Thus, our previous experiments with systematically varied synthetic CREs show that a sequence context composed of only two types of TFBSs strongly modulates the effects of CRX binding sites. However, it is unclear what kinds of interactions among CRX and NRL sites could account for such context-dependent activity.

To discover interactions among CRX and NRL sites that might explain context-dependent activity, we trained a model on the synthetic CRE dataset using MAVE-NN, a recently published neural network framework that is designed specifically to model data obtained from massively parallel functional assays [57]. A strength of MAVE-NN is that it deconvolves sequence-function relationships from the confounding effects of the measurement process, and thus for some model architectures parameters can be straightforwardly interpreted as additive and interaction effects of TFBSs. A key assumption of MAVE-NN is that each CRE

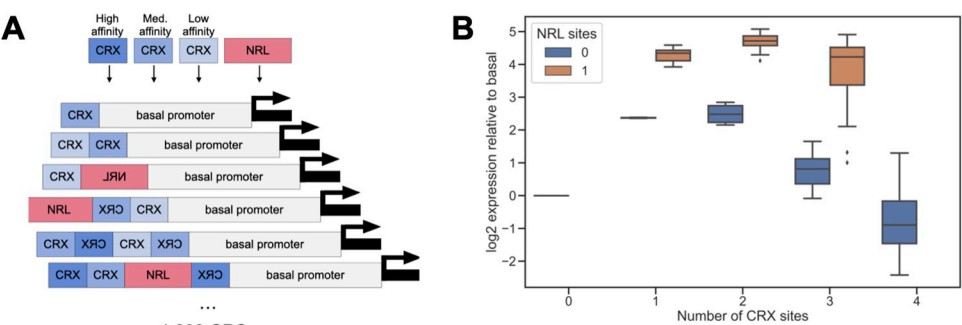

**Fig 1. Synthetic CREs sites reveal context-dependent effects of CRX and NRL sites.** (A) Design of synthetic CRE MPRA library reported in [13]. Combinations of CRX and NRL sites (up to four TFBSs) were cloned adjacent to either a *Rho* or a *Hsp68* basal promoter. TFBSs could be in either forward or reverse orientation. (B) MPRA activity (y-axis) of CREs composed only of high affinity CRX sites (blue) is consistently lower than that of CREs with high affinity CRX sites and one NRL site (orange), relative to the *Rho* basal promoter. Sequential addition of high affinity CRX first activates, then represses the *Rho* basal promoter. Plot shows a subset of the data reported in [13].

sequence has a well-defined *latent phenotype*, representing the intrinsic activity of the CRE that is indirectly captured by the experimental assay. The mapping from DNA sequence to latent phenotype is the relationship we seek to understand, but this relationship cannot be directly inferred from MPRA measurements due to non-linearities and noise in the measurement process. Therefore MAVE-NN simultaneously models (1) the mapping between DNA sequence and latent phenotype (called the "genotype-phenotype map"), and (2) the nonlinear relationship between the latent phenotype and the measured MPRA readout. To model the effects of the measurement process, MAVE-NN non-linearly maps the inferred latent phenotype to a prediction of the most probable measurement value. A skewed-t noise model is used to describe likely deviations from the most probable value. The parameters of the genotype-phenotype map can then be straightforwardly interpreted as additive and interaction effects between elements of the DNA sequence. MAVE-NN quantifies the performance of the models using an information theoretic measure called predictive information [57,62]. Predictive information is the mutual information between the inferred latent phenotype and the MPRA measurement, and it represents how well the model captures the relationship between a reporter gene's inferred intrinsic activity and its MPRA output.

We trained MAVE-NN models with different architectures to predict MPRA activities from sequence alone. We reasoned that due to the small number of TFBSs included in the synthetic CREs and the uniform spacing between them, additive models with or without interaction terms would capture most of the effects of CRX and NRL sites on reporter activity. We used predictive information to compare the performance of four different model architectures: (1) an additive model lacking interactions between TFBSs, (2) a nearest-neighbor model that only allows interactions between neighboring TFBSs, (3) a pairwise interaction model allowing interactions between all pairs of TFBSs regardless of spacing, and (4) a 'black box' multilayer perceptron model that makes no prior assumptions about the interactions between TFBSs. To train the models, the measurements from the *Rho* and *Hsp68* libraries were combined into one dataset, which was then randomly split among training (80%), validation (10%), and test (10%) sets. The expression from the *Hsp68* basal promoter was taken as the baseline level, which enabled us to specifically model the effect of the *Rho* promoter. All performance metrics were computed from the test set. Model parameters for analysis were taken from the best performing model out of multiple random initializations (see Methods).

Of the three model architectures that included additive and interaction terms, we found that the pairwise interaction model achieved the best overall performance, by both predictive information and Pearson correlation (Figs 2A and S2A). This model provided 1.8 bits of predictive information, roughly equivalent to an accurate three-way classification of CREs by activity. The predictive information of the pairwise model (1.82 bits) is approximately half that of the multilayer perceptron "black box" model (3.00 bits). The disparity in predictive information between the pairwise and black box models suggests that additional higher order interactions between TFBSs likely account for much of the unexplained activity of the synthetic CREs. However, this unexplained activity likely consists of small discrepancies between sequences with similar activities, because the pairwise model captured a substantial fraction of the variation in reporter activity (Figs 2B and 2C, $R^2$ = 0.889). To understand how additive and interaction effects of TFBSs might explain the context-dependent activity of CRX sites, we examined the parameters of the pairwise model. We estimated the parameter uncertainties using MAVE-NNs built-in functionality (S2B Fig, n = 20).

We found that the additive contributions of all TFBSs and the *Rho* promoter, averaged over all four possible positions in the synthetic CREs, were positive with the exception of the medium affinity CRX site in reverse orientation (Fig 2D). These positive effects are consistent with the roles of CRX and NRL as transcriptional activators. The average additive contribution

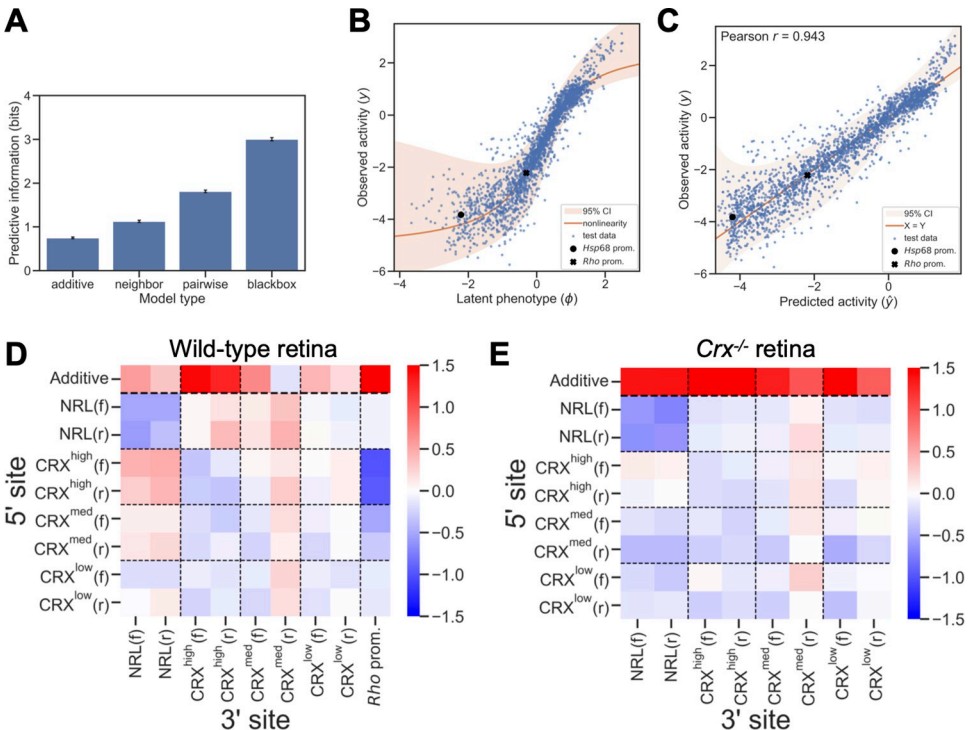

**Fig 2. A model of CRX and NRL-driven *cis*-regulatory activity in wild-type retina.** (A) The performance of different model architectures (measured as predictive information) fit to MPRA measurements of the CRX-NRL library in wild-type retina. Error bars indicate standard error. (B) The observed activity (y-axis) of test set sequences in wild-type retina compared to the latent phenotype (x-axis) inferred by the pairwise model. The non-linearity is the model mapping from latent phenotype to observed activity and intended to capture non-linear effects of the MPRA measurement process. (C) The observed activity (y-axis) of test set sequences in wild-type retina compared to the activity predicted by the pairwise model (x-axis). (D) Model parameters for additive and pairwise contributions of CRX and NRL sites and the *Rho* promoter to activity in wild-type retina, averaged across the four positions in synthetic CREs. For pairwise interactions, rows indicate the 5' site and columns indicate the 3' site. Forward and reverse orientation of the TFBS is indicated by (f) and (r). (E) Model parameters for additive and pairwise contributions of CRX and NRL sites to activity in *Crx*$^{-/-}$ retina, averaged across positions and spacings.

of CRX sites increases with site affinity, consistent with the additive contributions being driven by CRX binding. High affinity CRX sites have a stronger positive, additive effect than NRL sites, suggesting CRX is a stronger activator (Fig 2D). The *Rho* promoter had the strongest additive contribution, likely due to the presence of an NRL site and one high, one medium, and one low affinity CRX site [31,63]. The additive terms of the model reflect the expected effects of simple transcriptional activators whose probability of binding to a CRE is determined by the number and affinity of binding sites. However, these positive, additive terms alone do not account for the context-dependent effects of CRX binding sites observed in the data shown in Figs 1B and S1A.

Examining the interaction terms of the model, we observed a pattern of positive, heterotypic cooperativity between CRX and NRL sites, and negative homotypic interactions between binding sites for the same TF (Fig 2D). Negative homotypic interactions are strongest between NRL sites and between high affinity CRX sites, and they decrease with CRX site affinity, consistent with these interactions being determined by CRX binding. CRX sites also show a strong, affinity-dependent negative interaction with the *Rho* basal promoter, which likely reflects negative interactions with the promoter CRX sites. Negative homotypic interactions were especially strong between adjacent sites but can be observed at all distances in the

synthetic CREs (S2C Fig). Positive interactions between CRX and NRL sites also occur at all distances and depend on binding site affinity. The reverse orientation medium affinity CRX site was an outlier among CRX sites, exhibiting a slightly negative additive effect and positive interactions with all other TFBSs when averaged over CRE position (Fig 2D). Examining all pairwise parameters of the model shows that this trend is driven by a strong interaction between reverse medium affinity CRX sites with the adjacent 5' site. The unusual effect of this site was robust across multiple independently trained models (S2D Fig). It may reflect competitive binding by one of the other homeodomain TFs expressed in the retina, which may have a strong effect when bound immediately 3' of CRX or NRL [40]. Overall, the modeling results suggest that activating and inhibiting interactions between CRX and NRL are the primary determinants of the activity of CREs with binding sites for these two TFs. Because these interactions occur across all distances and not just at neighboring sites, the effects are likely due to multivalent interactions that involve co-factors and not only direct protein-protein contacts between CRX and NRL.

Taken together, the parameters of the pairwise interaction model reveal a *cis*-regulatory grammar that accounts for the observed context-dependent activity of CRX and NRL sites. Consistent with the known roles of CRX and NRL as transcriptional activators, sites for these TFs consistently make positive, independent contributions to activity. However, negative homotypic interactions reduce activation or lead to repression when multiple sites for the same TF are placed together. The repressive effect of negative homotypic interactions can be overcome by the strong heterotypic interactions between CRX and NRL. An important feature of this *cis*-regulatory grammar is that additive effects and interactions scale differently with the number of TFBSs. The independent, additive effects increase linearly with the number of TFBSs, while the interaction effects increase with the square of the number of TFBSs. These differences in scaling have a strong impact on CREs with multiple sites and explain why the replacement of a single binding site can convert a silencer to an enhancer (Fig 1B).

### Positive heterotypic interactions require CRX protein

We previously reported that many genomic and synthetic CREs with CRX binding sites either retain or gain activity in *Crx*$^{-/-}$ retina, despite the loss of CRX protein [13]. Activity in *Crx*$^{-/-}$ retina still requires intact CRX sites, indicating that another TF, likely the CRX ortholog OTX2, acts at these sites when CRX is absent. To examine how additive and pairwise interactions among TFBSs change in the absence of CRX, we trained a pairwise interaction model on prior data from the synthetic CRE library tested in *Crx*$^{-/-}$ retina. This library included only the *Rho* basal promoter. The *Crx*$^{-/-}$ model performed similarly to that trained on data from wild-type retina (2.21 bits of predictive information, $R^2$ = 0.900 for predicted versus observed activity, S2E-S2J Fig). In this model, additive effects of all TFBSs remained positive (Fig 2E), indicating that these CREs continue to be bound by transcriptional activators in *Crx*$^{-/-}$ retina. Unlike the model for wild-type retina, the additive contributions of CRX sites did not show a strong dependence on affinity. Negative homotypic interactions remain in *Crx*$^{-/-}$ retina, though they are attenuated for CRX sites. The altered effects of negative homotypic interactions in the *Crx*$^{-/-}$ model are evident in the MPRA data. The repressive effect of CRX sites is much weaker, while the homotypic effects of NRL sites are similar to those seen in wild-type retinas (S1B and S1C Fig, compare with Figs 1B and S1A). Notably, the positive interaction between CRX and NRL sites was absent, indicating that the interaction between these two sites depends specifically on CRX and NRL, and that other TFs that bind these sites in *Crx*$^{-/-}$ retina do not interact. Despite the loss of positive cooperativity between CRX and NRL sites, the model suggests that synthetic CREs in *Crx*$^{-/-}$ retina maintain or increase their activity due to

stronger additive contributions of lower affinity binding sites and a modest attenuation of negative homotypic interactions between CRX sites. Taken as a whole, the model suggests that cooperative interactions depend on the specific identities of the TFs involved, while the positive additive and negative homotypic effects hold more generally among TFs, though with varying effect sizes.

## Additional retinal TFs contribute to CRE activity

CRX and NRL are critical for establishing rod photoreceptor identity, and together they drive high expression of a number of key rod photoreceptor genes [35,39,45,47,64]. However, a cooperative interaction between CRX and NRL is not sufficient to fully explain the context-dependent effects of CRX sites in enhancers, because most CRX-bound enhancers do not contain a copy of the NRL motif [18,43]. To investigate how other TFBSs contribute to the activity of CREs with CRX sites, we designed a new library of 6,600 synthetic CREs. The library included TFBSs for CRX, NRL, and sites from three other motif families that occur in CRX-bound enhancers: NEUROD1, RORB, and motifs representing SP4 or MAZ [42,65–67]. These TFs are known to play roles in photoreceptor development and they are enriched at CRX-bound sites and accessible chromatin in rod photoreceptors [39,40,44]. We previously found that motifs for these TFs were enriched in CRX-bound enhancers relative to CRX-bound silencers [18], suggesting that their presence favors activation over repression. Therefore, unlike the CRX-NRL library, we expected most synthetic CREs in the new library to act as enhancers rather than silencers. The purpose of this library was to discover whether other TFBSs interact cooperatively with CRX sites, or whether they contribute independently to enhancer activity. Because the new library included more TFBSs than the CRX-NRL library, we could not exhaustively explore all possible combinations of TFBSs. We therefore fixed the lengths of the CREs at five sites and only tested TFBSs in the forward orientation. The NRL sites in this library differed from the consensus NRL site included in the CRX-NRL library. We selected high and medium affinity NRL sites that co-occur with NEUROD1, RORB, or MAZ sites in genomic enhancers [18]. Given these constraints, we designed synthetic CREs by systematically varying the TFBSs composition across the library. Each CRE included either two or three CRX sites and one or two sites for two additional TFs (Fig 3A). Synthetic CREs were cloned upstream of the *Rho* minimal promoter and tested by MPRA with three replicate transfections in explanted retinas (mean $R^2$ between replicates = 0.950, S3A Fig). We used the data to train different models and again found that the pairwise interaction model performed better than the additive or nearest-neighbor models. No additional performance was gained from the black box model (S3B and S3C Fig). The pairwise model captured most of the variance in CRE activity (Figs 3B and S3C, $R^2$ between predicted and observed expression = 0.985). We estimated parameter uncertainties using MAVE-NNs built-in functionality (n = 20, S3E Fig).

Examining the average additive effects of TFBSs in the pairwise interaction model, we found that higher affinity sites for NRL, NEUROD1, RORB, and MAZ contributed positively to activity, while lower affinity sites had weaker positive effects or negative effects on activity (Fig 3C). High affinity sites for NEUROD1 and RORB made especially strong additive contributions to activity. In contrast to the models above, the additive contributions of CRX sites in this model were negative. This is likely due to the design of the library, which only includes CREs with two or three CRX sites, making it difficult to deconvolve additive effects of individual CRX sites from the effects of negative homotypic interactions between CRX sites. The strong dependence of additive contributions on the sites' affinity for their cognate TFs is consistent with the effects arising from TF binding and occupancy. The interaction terms of the

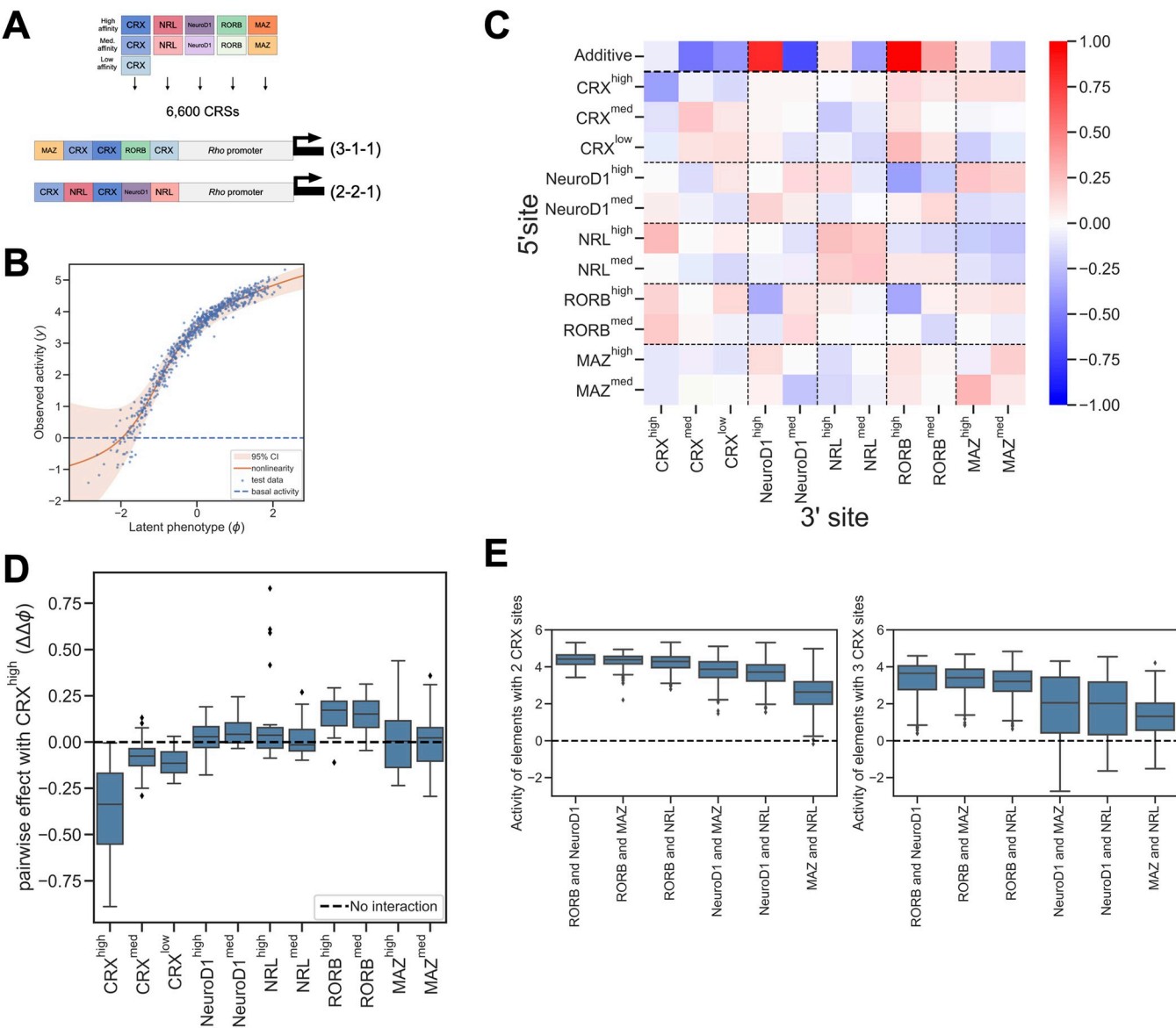

**Fig 3. A model of *cis*-regulatory activity driven by diverse TFBSs in wild-type mouse retina.** (A) Design of MPRA library of synthetic CREs with additional lineage-specific TFBSs. CREs contained five sites placed adjacent to the *Rho* basal promoter. Each CRE contained either three CRX sites and two sites for other TFs (3-1-1) or two CRX sites, two sites for another TF, and one site for a third TF (2-2-1). (B) Observed activity (y-axis) of test set sequences compared to the latent phenotype (x-axis) predicted by the pairwise model. (C) Model parameters representing additive and pairwise contributions of TFBSs averaged across positions. (D) Distribution of position-specific interactions with high-affinity CRX binding sites, broken down by partner TF. (E) MPRA activity of CREs with two (left) or three (right) CRX sites, grouped by TF identity. Each CRE contains sites for CRX and two other TFs. Activity is measured relative to the *Rho* basal promoter and basal activity is indicated by the dashed line.

model exhibit a mixture of moderate positive and negative effects that depend on the affinity and order of the TFBSs (Figs 3C and S3F).

To focus on interactions between other TFs and CRX, we examined the distribution of position-specific interactions of TFBSs with high affinity CRX sites within CREs (Fig 3D). While interactions varied somewhat with position, there were persistent effects for different TFBSs. As in the previous model, high affinity CRX sites exhibited strong negative homotypic interactions that attenuated as binding site affinity decreased. The strongest effects included

interactions with high affinity NRL sites, but only when those sites were adjacent to CRX (Figs 3D and S3F). The contrast with the CRX-NRL model (S2C Fig) could be due to differences in CRE structures between the two libraries or to differences in the NRL motif sequence (see Methods). Binding sites for RORB showed the most consistent positive interactions with CRX sites. The inferred parameters of medium affinity sites for NEUROD1, NRL, and MAZ sometimes differed from those of high affinity sites (Fig 3C). Medium affinity values were often lower than high affinity values, suggesting that this is a consequence of reduced TF occupancy. However, we cannot rule out the possibility that TFs from the same class with slightly different specificities may compete for binding to lower affinity sites.

Because each CRE in the library contained at least two other TFBSs in addition to CRX sites, we expected most CREs to act as enhancers rather than silencers, since these TFBSs are specifically enriched in genomic enhancers relative to silencers [18]. The MPRA results were consistent with our expectation (Fig 3E). We also found that the interaction effects inferred by the model corresponded with the observed activities. The model inferred that RORB sites interacted positively with CRX at most positions (Fig 3D), and CREs with RORB sites were consistently the most active in the library (Fig 3E). CREs without RORB were less active, though still above basal levels. Notably, both high affinity RORB and NEUROD1 sites are inferred by the model to have strong additive effects, but only RORB sites interact positively with CRX (Fig 3C). These results are consistent with a model in which effects of negative homotypic interactions between CRX sites can be overcome in two ways: (1) positive heterotypic interactions (CRX with RORB or a subset of NRL sites), and (2) independent, additive effects from TFBSs that don't strongly interact with CRX (MAZ, most NRL sites, and NEUROD1). In this model, CRE activity results from a quantitative balance of these effects. Supporting this hypothesis, CREs with only two CRX sites (Fig 3E, left panel) are more active than CREs with three CRX sites (Fig 3E, right panel). In both cases, CREs with RORB sites are more active than CREs lacking them. This suggests that positive heterotypic interactions balance negative homotypic interactions more effectively than the additive contributions of non-interacting TFs.

### Positive additive effects and negative homotypic interactions generalize to a second cell type

We asked whether the *cis*-regulatory grammar in other cell types included negative homotypic interactions balanced by positive heterotypic and independent effects. We trained pairwise interaction models on published MPRA data from a library of 4,966 synthetic CREs composed of binding sites for twelve liver-specific TFs, tested in HepG2 cells and mouse liver [68]. CREs were composed of homotypic or heterotypic arrangements of up to twelve consensus binding sites placed into two different neutral sequence templates.

Averaging measurements across replicates and splitting the data into training, validation, and test sets led to poor model performance ($< 1$ bit of predictive information). As an alternative approach, we treated each of the replicates as a separate set of measurements, and split the combined dataset composed of measurements for each combination of replicate and sequence randomly into training, validation, and test sets. We were able to fit well-performing models (predictive information $> 1$ bit, Pearson R 0.87–0.92) for both the *in vivo* mouse liver data (S4A and S4B Fig) and HepG2 cell data (S5A and S5B Fig). Estimates of parameter uncertainties suggest that parameters for the different TFs are reliably distinguished from each other (S4C and S5C Figs)

We examined the additive and pairwise interaction terms of the models averaged across CRE positions. The inferred interactions of mouse liver and HepG2 models were similar,

despite significant differences between the *in vivo* and *in vitro* cellular contexts of the MPRA datasets (S4D-S4F and S5D-S5F Figs). Notably, most TFBSs exhibit negative homotypic interactions, with sites for HNF1A showing the strongest effect (seen along the heatmap diagonal of S4D and S5D Figs, see also S4F and S5F Figs). The negative homotypic interaction between HNF1A sites is consistent with observation in the MPRA data that clusters of HNF1A sites quickly reach saturation, showing little gain in activity when the number of sites is increased from four to eight (see Fig 2A in reference [68]). The additive effects of most TFs were positive, though often only weakly positive (S4E and S5E Figs). Notably the sites with the strongest additive effects, HNF1A and XBP1, also exhibited the strongest negative homotypic interactions. This may indicate that negative homotypic interactions are mediated via the transcriptional effector domains of these TFs. More generally, the pattern of positive additive contributions and negative homotypic interactions inferred by the model are consistent with the experimental observation that heterotypic clusters of TFBSs drive stronger expression than homotypic clusters in this MPRA library (see Fig 3 in reference [68]). Together, these results suggest that the pattern of positive additive contributions and negative homotypic interactions may be a recurring feature of transcriptional activators across cell types and species.

### Balance between positive and negative interactions can explain context-dependent effects of binding sites for transcriptional activators

Our models suggest that sites for transcriptional activators like CRX and NRL can show context dependent activity that results from a quantitative balance between negative homotypic interactions, positive heterotypic interactions, and positive, independent effects of individual TFBSs. These effects could lead to context-dependent silencing and activation without the need for specific repressor TFBSs. The results presented above suggest that some transcriptional activators self-inhibit when present at higher occupancy on a CRE. The negative effects of self-inhibition can be overcome in two ways, via positive cooperativity with a different TF, or by the non-cooperative action of a diverse collection of TFs. Under this model, the TFBS composition at enhancers and silencers shifts the balance between these effects in favor of either activation or repression. At enhancers, positive cooperativity and the independent contributions of diverse activator TFBSs outweigh the effects of negative homotypic interactions, while at silencers negative homotypic interactions predominate.

To illustrate this hypothesized *cis*-regulatory grammar, we implemented a simplified model that expresses CRE activity as the sum between positive and negative contributions of activator TFBSs. This model has no fit parameters and is intended to explore the effects of positive heterotypic interactions and non-cooperative, additive effects. It thus ignores order and orientation effects, which are clearly still relevant, as seen in parameters of the MAVE-NN models (S2C, S2J, and S3F Figs). While there is a significant proportion of variability in CRE activity that cannot be explained by binding site composition alone (Fig 1B), the non-monotonic relationship we observe between the number of CRX sites and CRE activity (Fig 1B), which this model recapitulates, is the largest contributor to CRE activity. Binding site composition is thus the primary factor distinguishing strong enhancers from silencers composed of CRX and NRL sites, as measured in our assay.

In the simplified model, we assume that CRE activity is the sum of (1) positive, additive contributions from sites for transcriptional activators, (2) positive cooperativity between sites for different TFs, and (3) negative interactions between sites for the same TF. For CREs composed only of sites for two different TFs, as in Fig 1, this sum is

$$A = \alpha_x x + \alpha_y y + \beta_{xy} xy - \frac{\gamma_x x(x-1) + \gamma_y y(y-1)}{2} \tag{1}$$

where *A* is activity of a CRE, *x* is the number of sites for the first TF, *y* is the number of sites for the second TF, and *α, β, γ* are weights reflecting the relative strength of each contribution to activity. The first two terms represent the additive contribution of each TFBS, the third term represents positive cooperativity between all pairs of sites for different TFBSs, and the final term represents negative interactions between all pairs of sites for the same TF.

To illustrate how heterotypic interactions balance negative homotypic interactions, we calculated the expected activities of all possible CREs composed of CRX and NRL sites, up to a total of four sites (Fig 4A). We did not fit any parameters, and instead made the simplifying assumption that the relative strengths of the different terms in Eq 1 are similar by setting all weights equal to 1. The activities simulated with the model accurately recapitulate the patterns of expression observed in the CRX-NRL library. Starting with the basal promoter alone

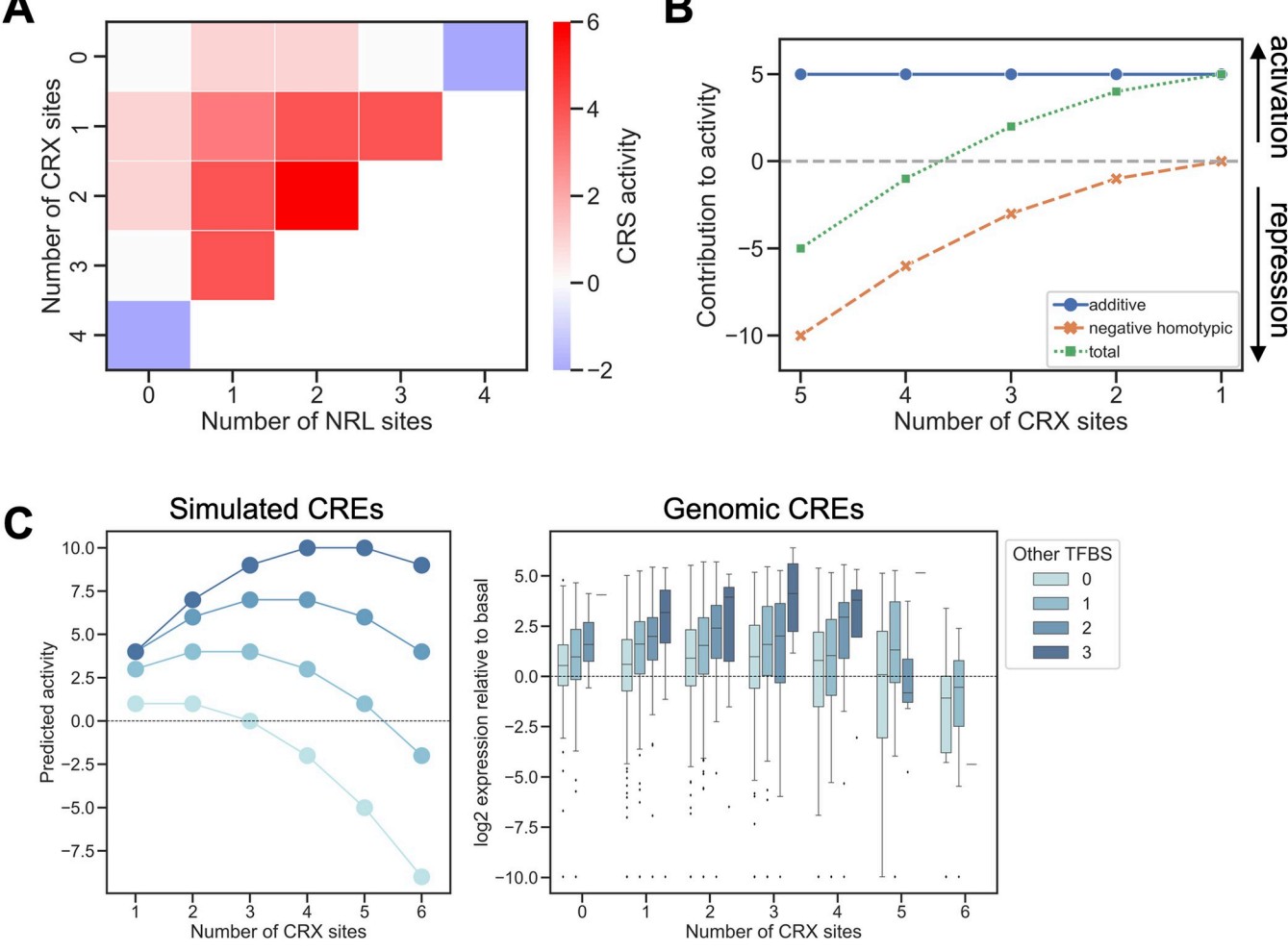

**Fig 4. Simplified balance model of context-dependent effects of binding sites for transcriptional activators.** (A) Simulated CRE activities calculated by Eq 1 for sequences with up to four TFBSs for CRX or NRL. Stepwise addition of sites for a single TF first increase then decrease activity. The first two columns show predicted expression of CREs with only CRX sites or with CRX sites plus one NRL site. Compare with the measured values in Fig 1B. (B) Model of CREs with five TFBSs shows how TF diversity reduces negative homotypic interactions and increases CRE activity. As CRX sites are replaced with sites for different TFs, TF diversity increases (x-axis) and the number of negative homotypic interactions decreases (orange crosses) and the overall CRE activity increases (blue squares). The total additive contribution of TFBSs (green circles) is equal to the total number of TFBSs and remains constant. (C) Simulated CRE activities calculated by Eq 1 (left panel) recapitulate trends in MPRA activity of genomic CREs (right panel). Increasing numbers of CRX sites first increase then decrease activity, but the presence of TFBSs for NRL, NEUROD1, RORB, or MAZ ('Other TFBSs') increases both predicted and measured CRE activity. Dashed lines indicate basal activity.

(Fig 4A, indicated by zero), a stepwise addition of sites for a single TF leads first to an increase and then a decrease in activity (leftmost column or top row in Fig 4A, compare with data Figs 1B and S1A). The highest activities are obtained from CREs with combinations of sites from both TFs. In this simplified model, a CRE with four CRX sites is repressive. Replacing one of those sites with an NRL site converts the CRE from a silencer to an enhancer, an effect also observed in the data (compare Figs 4A and 1B). As a further test of the simplified model, we used Eq 1 to calculate predicted activities for the entire CRX-NRL library. Comparing these predictions with the measured activities, we obtained a Spearman correlation of 0.76. In the *Crx*$^{-/-}$ dataset, this correlation reduced to 0.18 due to the loss of homotypic interactions between CRX sites. For the longer, more complex synthetic CREs composed of liver TFBSs (Figs S4 and S5 and reference [68]), the simple model predicts the rank order of CRE activity less well (HepG2 Spearman correlation = 0.43, Mouse Spearman correlation = 0.53). This is likely due to a larger influence of spacing, order, and orientation in longer synthetic CREs. While our model relies on simplifying assumptions that do not fully hold *in vivo*, it successfully recapitulates major trends in the data.

To illustrate the second mechanism of balancing negative homotypic interactions, via independent positive effects of diverse TFBSs, we implemented the simple model without cooperative interactions. The intuition is that, for a CRE composed of a given number of sites, greater TFBS diversity reduces the total number of negative homotypic interactions present. Thus, TFBS diversity leads to activation even in the absence of strong cooperativity. In this regime, the independent positive effects of each TFBS predominate, as our MAVE-NN model suggests for CREs with NEUROD1 and MAZ sites (Fig 3C and 3D). To capture this mechanism with our simplified model, we assumed that (1) bound TF activators always make positive, independent contributions to activity and (2) all TFs engage in negative homotypic interactions. We calculated the sums of positive and negative effects for CREs with five total TFBSs, but different numbers of CRX sites. In these simulated CREs, the total additive contribution is constant and equal to the total number of TFBSs (Fig 4B, blue circles). As the TFBS diversity increases, the total number of negative homotypic is reduced (Fig 4B, orange crosses). A CRE with five CRX sites is therefore highly repressive, while replacing some CRX sites with different TFBSs increases the activity of the CRE (Fig 4B, green squares). This simplified model demonstrates how strong activity can be achieved by the independent effects of diverse TFBSs, even in the absence of cooperative interactions. This model suggests an explanation of our prior observation that strong enhancers have more diverse TFBSs than silencers [18].

We tested the ability of the simple interaction model to explain patterns of activity among genomic CREs. We used Eq 1 to simulate the activities of CREs with up to 6 CRX binding sites and up to 3 sites for non-CRX TFs (Fig 4C, left panel). As with the simpler simulation (Fig 4A), the model predicts that increasing the number of CRX sites eventually leads to repression, which is counteracted by the presence of non-CRX TFBSs. We then compared the activity pattern of the simulated CREs to the activities of genomic CRX-bound CREs that we previously assayed by MPRA in retinal explants (Fig 4C, right panel) [18]. The simulated data recapitulated the major trends in the MPRA data, showing that a simple model of positive and negative interactions captures important factors that determine the activity of genomic CREs.

## Discussion

Because the effects of a TFBS often strongly depend on local sequence, the activity of *cis*-regulatory DNA is not a simple function of its TFBS composition. To accurately predict the activities of *cis*-regulatory sequences and the effects of genetic variants that occur within them, we need models of *cis*-regulatory grammar that accurately account for the influence of sequence

context. The contribution of a TFBS to the activity of a CRE can vary due to post-translational modifications of TFs [69,70], the presence of other co-bound factors [5,6,9,11,12,19,29,71–74], and binding by different TFs with similar sequence specificities [10,75,76]. Our results suggest that context dependence can also be determined by the overall balance between the independent and interaction effects of individual sites for transcriptional activators. At the core of this model is a distinction between additive, independent effects of individual TF molecules and effects of interactions between molecules. In the case of CRX, the independent and interaction effects influence *cis*-regulatory activity in opposite directions, with CRX molecules independently contributing to activation while engaging in repressive homotypic interactions with one another. The independent, activating effects scale linearly with the number of binding sites, while the number of repressive homotypic interactions scales with the square of binding site number. As the number of binding sites in a CRE increases, negative homotypic interactions grow faster than the activating effects of individual binding sites. As a result, sequences with many CRX sites are likely to act as silencers, a pattern that we observe with both synthetic CREs and genomic CRX-bound sequences [13,18].

In this model, there are two ways in which other TFs influence whether a CRX-bound CRE will activate or repress transcription. First, a TF like NRL may form positive cooperative interactions with CRX. Because positive interactions also scale with the square of the number of binding sites, the presence of even one or two sites for a cooperating TF can shift the balance towards activation (Fig 4A). Second, multiple independent, additive contributions from other TFs can outweigh negative homotypic effects, even in the absence of positive cooperativity (Fig 4B). Taken together, these two means of countering negative homotypic interactions imply that a more diverse set of TFBSs within a CRE will minimize negative homotypic interactions and lead to stronger enhancer activity.

Our model is supported by previous observations that we made of genomic CRX-bound CREs. We previously reported that genes near sites that are co-bound by both CRX and NRL are more highly expressed than genes near regions bound by CRX alone [13]. Homotypic clusters of CRX sites in genomic sequences are often repressive, and CRX-bound silencers tend to have more CRX sites than CRX-bound enhancers. When CRX sites in silencers are abolished, repressive activity is lost [13,18,43]. Similar to the results with synthetic CREs (Fig 1B), genomic CREs show a quantitative correlation between the number of CRX sites and the tendency to be repressive, which supports our model of negative homotypic interactions (Fig 4C) [13]. The model presented here also suggests an explanation for our prior observation that CRX-bound enhancers often have more diverse TFBSs than CRX-bound silencers [18]. Genomic sequences that act as silencers often contain clusters of multiple CRX sites and few sites for other TFs, while genomic enhancers tend to contain sites for a variety of photoreceptor TFs [18]. We previously reported that a simple measure of TFBS diversity could partially classify enhancers from silencers [18]. Our MAVE-NN models suggest an explanation for this phenomenon: enhancers have more diverse TFBSs because diversity allows the additive contributions of TFBSs to outweigh negative homotypic interactions. This contrasts with low-diversity silencers, which our model suggests are dominated by negative homotypic interactions. TFBS diversity is a feature of enhancers in other cell types, where a similar balance between positive additive effects and negative homotypic interactions may occur [68,77]. Because the weights we used to interpret the models trained on the CRX-NRL datasets are automatically adjusted to assign an activity of 0 to the *Hsp68* promoter without any added CRX or NRL binding sites, we are confident that the signs of the additive and interacting terms in these models do in fact reflect a distinction between activation and repression. However, we do not have such a baseline in the other datasets with the result that we cannot definitively ascribe the effects of the TF binding sites in that library to activation or repression.

A key prediction of our model is that negative homotypic interactions strongly influence the context-dependent effects of binding sites for transcriptional activators. The existence of such interactions in photoreceptors is supported by data from both synthetic CREs and genomic sequences. Sequential addition of CRX or NRL binding sites upstream of the *Rho* basal promoter first increases, then decreases transcription, sometimes below basal levels (Figs 1B and S1) [13]. Genomic CRX-bound sequences that act as silencers when measured by MPRA have more copies of the CRX motif than sequences that act as enhancers [13,18,43]. We have shown that this silencing activity depends on both CRX motifs and CRX protein [13]. Similar negative homotypic effects have been reported for several TFs, including liver-specific factors [68], yeast Gcn4 [78], pluripotency TFs [16,51], and Sp3 [79]. These findings suggest some TFs may self-inhibit or recruit repressors when present at high CRE occupancy [20,80,81]. A clear example of such a mechanism is the homeodomain TF WUSCHEL, which activates transcription as a monomer at low concentration but forms repressive dimers at higher concentration [82,83]. Such effects may be common among homeodomain TFs, which are enriched in transcriptional effector domains, including the CRX effector domain, that exhibit the ability to both activate and repress [84]. Our model of context dependency suggests that the balance between positive effects and negative homotypic interactions can account for the dual activities of some TFs, without the need for dedicated repressors.

Homotypic clusters of TFBSs are a common feature of eukaryotic CREs and their functional effects vary [85–88]. In *Drosophila*, homotypic clusters of Zelda but not Bicoid binding sites drive expression in developing embryos, showing that TF-specific properties determine the effect of homotypic clusters [89]. Similarly, homotypic clusters of yeast TFBSs can have positive or negative effects, depending on the identity of the TF [90]. In a recent MPRA study of binding sites for eighteen liver-associated TF, CRE activity increased with the copy number of homotypic TFBSs for eight TFs, while activity decreased for six TFs [91]. TFBS orientation affected the impact of homotypic clusters, showing that the repressive effects of homotypic interactions can depend on how TFBSs are configured. TFBS affinity also matters: homotypic clusters of low-affinity binding sites can produce specific TF binding at functional levels, thereby achieving discrimination among TFs with similar binding specificities [92,93]. In our model of CREs with CRX binding sites, TFBSs orientation generally has a minor impact on homotypic negative interactions, while affinity has a strong effect, with the highest affinity sites producing the strongest homotypic interactions (Fig 2D). Based on our results and the studies cited above, negative homotypic interactions may be common among many classes of TFs, but the impact of these negative interactions on CRE activity clearly depend on specific properties of TFs and TFBSs.

There are several potential mechanisms that could cause the negative homotypic interactions inferred by our model. CRX may preferentially interact with other TFs at low or medium CRE occupancy, while at high occupancy CRX may form repressive homodimers, as has been observed for at least one homeodomain TF [82,83]. High local occupancy may also lead to cooperative recruitment of co-repressors known to interact with CRX [94–96]. An alternative model is 'TF sharing', which occurs when multiple binding sites in the same CRE compete for a limited local pool of TFs. Thermodynamic modeling shows that under these conditions, an additional TFBS can reduce CRE activity [78]. Kinetic mechanisms could also potentially account for repressive effects of multiple TF activators. Some combinations of bound TFs and co-factors may affect the kinetics of specific steps of transcription initiation in ways that reduce basal levels of transcription [97] or lead to negative kinetic synergy [98].

Our results in *Crx*[-/-] retina suggest that the repressive effects of homotypic clusters of CRX sites may be cell type-specific. We observed that some CREs with homotypic clusters of CRX sites act as silencers in wild-type retina, but convert to enhancers in *Crx*[-/-] retina [13,99]. In

both genetic contexts, CRX binding sites are causal. Abolishing the homotypic clusters of CRX sites leads to loss of repression in wild-type retina and loss of activation in $Crx^{-/-}$ retina. This indicates that some homotypic clusters of CRX sites are pleiotropic, acting repressively when bound by CRX, but activating transcription when bound by another TF. That TF is likely OTX2, an ortholog of CRX with nearly identical binding specificity, and which is co-expressed in photoreceptors and bipolar cells [100]. OTX2 may bind homotypic clusters of 'CRX' sites in the absence of CRX ($Crx^{-/-}$ retina) or when CRX levels are low (bipolar cells). Our results suggest a mechanism by which homotypic clusters of CRX sites could have pleiotropic, cell type-specific functions: In photoreceptors, these clusters are bound by CRX and are repressive, while in bipolar cells, the same clusters may be bound by OTX2 and become activating. The MAVE-NN model indicates that, at least in $Crx^{-/-}$ retina, this conversion is due to a shift in the balance between negative homotypic interactions and additive contributions. Negative homotypic interactions between TFBSs persist in the absence of CRX, but these are outweighed by stronger additive contributions (Fig 2E). Our experimental results and MAVE-NN models show that the effects of local sequence context on TFBS activity can change when different TFs bind the same sites in different cellular environments.

While our results suggest that the balance between negative homotypic interactions and positive heterotypic and independent effects play an important role, more complex factors also play a role in the CRX-directed *cis*-regulatory grammar. Our pairwise interaction models show that spacing, orientation, and binding site sequence affect CRE activity in complicated ways that are not captured by our simple model (S2C, S2J, and S3F Figs). However, negative homotypic interactions may be a common feature of certain classes of transcriptional activators, and such interactions may explain why many TFs frequently play dual roles as activators and repressors.

## Methods

### Ethics statement

Animal procedures were performed in accordance with the Washington University in St. Louis Institutional Animal Care and Use Committee under approved protocol #D16-00245.

### Model fitting

For previously published data, we used processed data taken directly from published supplemental data files. For the CRX + NRL library (n = 1,299), binding site arrangements and MPRA activities were extracted from Database S3 of [13]. For the liver TF library (n = 4,966), data was taken from Supplementary Table 4 of [68]. For the library with CRX, NEUROD1, NRL, RORB, and MAZ sites (CDNRM library, n = 6,600), the MPRA experiment was performed as described below. Data files are described below under **Data availability**. To encode arrangements of TFBSs as input sequences for MAVE-NN, we used single letters to represent each type of binding site. To create input sequences of uniform length for the CRX + NRL library, dummy binding sites labeled "O" were prepended to each arrangement to render all CREs four sites long, and an additional letter indicating the basal promoter (*Rho* or *Hsp68*) was then appended to the end. Because the binding sites in the library of reference [68] did not occur at consistent locations, the sequences were segmented into 9 bp bins. If the 5' end of a binding site fell within a bin, a letter encoding the specific TF it binds was included in the corresponding location in the binding site sequence. Otherwise, a letter encoding the background sequence (B for the more active, b for the less active) was added as a spacer. Models were trained using mave-nn package version 1.01 until convergence on the processed data. Training

**Table 1. Hyperparameters used to fit models.**

| | |
|---|---|
| Learning rate | 0.001 |
| Number of Epochs | 1000 |
| Batch size | 200 |
| Early stopping patience | 30 |

histories are shown in S6A-S6C Fig. The hyperparameters are given in Table 1. They were chosen by manually adjusting the MAVE-NN defaults to obtain the best fit to the validation set. The models were specified using the Skewed-T GE noise model with a heteroskedasticity order of 2. We used the consensus gauge with basal *Hsp68* as the consensus sequence to obtain parameters from the models trained on CRX-NRL data in wild-type retina, and the uniform gauge for the remaining models. To ensure consistent training outcomes, we trained each model from multiple random initializations (25 for the CRX + NRL library in wild-type retina, 20 for the library in *Crx*$^{-/-}$ retina, 50 for the CDNRM library, and 20 for the liver TF library), with the numbers chosen to achieve maximum performance and reproducibility. We picked the best-performing model of each type for further evaluation. Model performance was evaluated by cross-validation with an 80-10-10 percent training-validation-test set split. The measurements were split randomly between sets and the same split was used for all random initializations. To estimate parameter uncertainty, we used MAVE-NN's built-in bootstrap function [57]. The function works as follows: beginning with a model trained on the original data, MAVE-NN simulates n datasets, on which new models are trained. Parameter uncertainties are then calculated from the n models trained on the simulated datasets. The datasets are generated by simulating a measurement for each CRE in the original dataset. CRE sequences in the simulated datasets are thereby kept fixed, with only the measurements treated as stochastic. This approach avoids the potential issue of parameter non-identifiability in models trained on simulated datasets lacking a substantial subset of the original sequences. We used the best-performing pairwise model for each dataset to generate 20 simulated datasets using this procedure.

## Simulating CRE activity with a simplified model

We set all constants in Eq 1 to one and used it to calculate activity of simulated CREs composed of different numbers of binding sites. Genomic CREs shown in Fig 4C were taken from reference [18]. The number of CRX and non-CRX binding sites in each genomic CREs was taken from the predicted occupancy calculations of reference [18].

## CDNRM library design

We designed a library of 6600 synthetic CREs composed of combinations of binding sites for CRX, NRL, NEUROD1, RORB, and MAZ. The library was designed to vary TFBS diversity around CRX sites. It contained all possible arrangements of either 3 sites for CRX and 1 site each for two other TFs (3-1-1 sequences); or 2 sites for CRX, 2 sites for a second TF, and 1 site for a third TF (2-2-1 sequences). CRX sites in a CRE were either all high affinity or a mixture of affinities. TFBS orientation was held constant. High, medium and low affinity CRX sites were those used in the CRX + NRL library [13]. To ensure that we picked motif sequences that are functional in at least one context, we picked high and medium affinity NRL, NEUROD1, RORB, and MAZ sites from genomic strong enhancers that lose some activity when the corresponding motif is deleted [18]. The high and medium affinity NRL sites in the CDNRM library differ from the consensus site in the CRX + NRL library. They were chosen because they co-

occur in genomic enhancers with other TFBSs tested in the CDNRM library. Binding site sequences were padded to make all motifs 12 bp, then a constant buffer sequence was added (AGCTAC<motif>GT) to create a 20 bp "building block" that maintains helical spacing when sites were combined, similar to our procedure for prior libraries of synthetic CREs [13,16,51]. The 12 bp motifs used with the core motif underlined, are: high affinity CRX, TG<u>CTAATCCCAC</u>; medium affinity CRX, TG<u>CTAAGCCAA</u>C; low affinity CRX, TG<u>CTGATTCAA</u>C; high affinity NRL: <u>AATTTGCTGACC</u>; medium affinity NRL, <u>GGCCTGCTGACC</u>; high affinity NEUROD1, C<u>AACAGATGGT</u>A; medium affinity NeuroD1, C<u>GGCAGGTGGT</u>A; high affinity RORB, <u>AATTAGGTCACT</u>; medium affinity RORB, <u>ATCTGGGTCAGT</u>; high affinity MAZ, <u>GGGGGAGGGGGG</u>; medium affinity MAZ, <u>GCGGGCGGGGGG</u>.

## MPRA library cloning

Synthetic CREs were each represented in the library with 3 unique barcodes. As standards, the library included 20 genomic sequences taken from [18] that span the dynamic range of the MPRA and 150 scrambled sequences as negative controls. The *Rho* basal promoter was tagged with 90 barcodes to ensure precise measurement of basal levels. Barcoded CREs were synthesized as two sub-libraries on a single chip using custom oligonucleotide synthesis from Agilent Technologies. The oligonucleotide libraries were cloned as previously described [18]. Briefly, we amplified oligos using either primer pairs MO563 (GTAGCGTCTGTCCGTGAATT) and MO564 (CTGTAGTAGTAGTTGGCGGC) or RZFP3 (TCTAGACTGCGGCTCGAATT) and RZFP4 (AGATCTAATGCATACGCGGC), and cloned them into the vector pJK03 (AddGene #173,490). The rod-specific *Rho* promoter, the *DsRed* reporter gene, and a multiplexing barcode (mBC) was cloned between the synthetic sequence and the cBC. One sub-library was assigned mBC TAGTAACGG, the other was assigned CCTACTAGT. The final plasmid libraries were pooled at equimolar concentrations.

## Retinal explant electroporation

Electroporations into retinal explants into P0 CD-1 mice and RNA extractions were performed as described previously [13,18,43,63]. We performed three replicate electroporations. cDNA and the input plasmid pool was sequenced on the Illumina NextSeq platform. We obtained an average sequencing depth of >675 reads per barcode.

## MPRA data processing

All sequencing reads were processed regardless of quality score. Sequencing reads were filtered to retain only perfect barcode matches. After filtering we retained 95% of sequencing reads. Barcodes with fewer than 50 reads in the plasmid pool were considered missing and removed. Barcode read counts were normalized by total sample reads to compute reads per million for each barcode. MPRA activity scores for each replicate were calculated by dividing RNA by DNA values, averaging across barcodes for each CRE, then normalizing to the activity of the basal promoter [18]. Replicates were averaged and the $\log_2$ transformed values were used for model training.

## Supporting information

**S1 Fig. Increasing NRL sites reduces MPRA activity in synthetic CREs.** Synthetic CREs composed only of NRL sites show an increase, then a decrease in activity relative to the *Rho* basal promoter as the number of sites is increased. Plot shows a subset of the data reported in

[13].
(TIF)

**S2 Fig. A model of CRX and NRL-driven cis-regulatory activity in *Crx*<sup>-/-</sup> retina.** (A) The performance of different model architectures measured by Pearson correlation coefficients, wild-type retina. (B) Estimates of model parameter uncertainties for additive (left) and pairwise interaction (right) parameters, generated by the built-in MAVE-NN function to estimate uncertainties. Interaction parameters are ordered by rank for visualization. (C) Model parameters for position-specific pairwise contributions of CRX and NRL sites in wild-type retina. Forward and reverse orientation of binding sites is indicated by capital or lower case letter. CRX sites are either high (C or c), medium (M or m), or low (L or l) affinity. NRL sites are labeled N or n and the *Rho* promoter is labeled R. There are no model parameters for *Hsp68* (H, used as the overall basal sequence) or the placeholder site _ used to equalize the lengths of input sequences. See methods for details. (D) Anomalous activity of reverse medium affinity CRX site, compared to the forward site. Additive parameters shown for independently trained pairwise models initialized from different random seeds (n = 8). (E, F) Performance of different model architectures fit to MPRA measurements of the CRX-NRL library in *Crx*<sup>-/-</sup> retina. (G) Estimates of model parameter uncertainties for additive (left) and pairwise interaction (right) parameters in *Crx*<sup>-/-</sup> retina. (H) Observed activity (y-axis) of test set sequences vs the latent phenotype inferred by the pairwise model of *Crx*<sup>-/-</sup> retina. (I) Observed activity (y-axis) of test set sequences vs predicted activity of the pairwise model of *Crx*<sup>-/-</sup> retina. (J) Position-specific pairwise contributions of CRX and NRL sites to activity in *Crx*<sup>-/-</sup> retina.
(TIF)

**S3 Fig. A model of *cis*-regulatory activity driven by diverse TF binding sites in wild-type retina.** (A) Reproducibility of MPRA measurements across three replicates. (B,C) Performance of modes fit to measurements of the MPRA library of CREs composed of five TFBSs, expressed in terms of predictive information and Pearson correlation. (D) Observed activity (y-axis) of test set sequences compared to the activity predicted by the pairwise model (x-axis). (E) Estimates of model parameter uncertainties for additive (left) and pairwise interaction (right) parameters. Interaction parameters are ordered by rank for visualization. (F) Position-specific pairwise contributions of diverse TF binding sites. Capital and lowercase letters represent high and medium affinity sites for CRX (C), NEUROD1 (D), NRL (N), RORB (R), and MAZ (M). Low affinity CRX sites are represented by x. Dashed boxes indicate strong cooperative interactions of high affinity NRL sites with CRX sites.
(TIF)

**S4 Fig. A model of *cis*-regulatory activity driven by diverse TFBSs in mouse liver.** (A) Observed activity (y-axis) of test set sequences compared to the latent phenotype (x-axis) predicted by the pairwise model. (B) Observed activity (y-axis) of test set sequences compared to the activity predicted by the pairwise model (x-axis). (C) Mean homotypic interaction contribution of each TF plotted against mean additive contribution. Error bars are standard deviations across 20 simulated replicates generated by MAVE-NN's built-in parameter uncertainty estimation function. (D) Model parameters representing pairwise contributions of TFBSs averaged across positions. (E) Distributions over simulated replicates of mean additive contribution from each TF averaged across positions. (F) Distributions over simulated replicates of mean homotypic interaction contribution from each TF averaged across pairs of positions.
(TIF)

**S5 Fig. A model of *cis*-regulatory activity driven by diverse TFBSs in HepG2 cells.** (A) Observed activity (y-axis) of test set sequences compared to the latent phenotype (x-axis)

predicted by the pairwise model. (B) Observed activity (y-axis) of test set sequences compared to the activity predicted by the pairwise model (x-axis). (C) Mean homotypic interaction contribution of each TF plotted against mean additive contribution. Error bars are standard deviations across 20 simulated replicates. (D) Model parameters representing pairwise contributions of TFBSs averaged across positions. (E) Distributions over simulated replicates of mean additive contribution from each TF averaged across positions. (F) Distributions over simulated replicates of mean homotypic interaction contribution from each TF averaged across pairs of positions.
(TIF)

**S6 Fig. Training history of pairwise models.** Training and validation set loss by epoch for MAVE-NN pairwise interaction models for (A) CRX-NRL library in wild type retina, (B) CRX-NRL library in *Crx*<sup>-/-</sup> retina, and (C) CDNRM library in wild-type retina.
(TIF)

## Author Contributions

**Conceptualization:** Kaiser J. Loell, Ryan Z. Friedman, Barak A. Cohen, Michael A. White.

**Formal analysis:** Kaiser J. Loell, Michael A. White.

**Funding acquisition:** Barak A. Cohen, Michael A. White.

**Investigation:** Kaiser J. Loell, Ryan Z. Friedman, Connie A. Myers, Joseph C. Corbo, Barak A. Cohen.

**Methodology:** Kaiser J. Loell, Ryan Z. Friedman, Joseph C. Corbo.

**Project administration:** Barak A. Cohen, Michael A. White.

**Supervision:** Barak A. Cohen, Michael A. White.

**Visualization:** Kaiser J. Loell, Michael A. White.

**Writing – original draft:** Kaiser J. Loell, Michael A. White.

**Writing – review & editing:** Ryan Z. Friedman, Joseph C. Corbo, Barak A. Cohen, Michael A. White.

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
