## [Decision Letter · Decision Letter 0]

26 Apr 2023

Dear Dr. White,

Thank you very much for submitting your manuscript "Transcription factor interactions explain the context-dependent activity of CRX binding sites" for consideration at PLOS Computational Biology.

As with all papers reviewed by the journal, your manuscript was reviewed by members of the editorial board and by several independent reviewers. In light of the reviews (below this email), we would like to invite the resubmission of a significantly-revised version that takes into account the reviewers' comments.

We cannot make any decision about publication until we have seen the revised manuscript and your response to the reviewers' comments. Your revised manuscript is also likely to be sent to reviewers for further evaluation.

Sincerely,

Jie Liu

Academic Editor

PLOS Computational Biology

Jian Ma

Section Editor

PLOS Computational Biology

Reviewer's Responses to Questions

**Comments to the Authors:**

Reviewer #1: This manuscript addresses an interesting question regarding what sequence context drives enhancer activity and which sequence content occurs within silencers. They use a neural network called MAVE-NN to mine MPRA data tested in photoreceptor cells. The model has good predictive ability and finds that enhancers contain more pairwise combinations of motifs while silencers contain homotypic clusters of CRX sites. This is a great paper that provides some interesting insights into gene regulation and what makes something an enhancer or a silencer. I very much enjoyed reading this paper. I have some minor comments.

• Can the authors speculate on why homotypic clusters could lead to silencer activity?

• Are there other examples of homotypic clusters leading to silencer activity, and are there examples of homotypic clusters leading to activation? Is this finding transcription factor specific, for example does it only apply to CRX, or certain types of transcription factors? Are there other examples of transcription factors that act as both activators and repressors where homotypic clusters are silencers or the opposite where such transcription factor clusters act as enhancers?

• Are these findings cell type-specific? For example, within another cell context where CRX is expressed, do the authors think they’d find clusters of homotypic sites would be activating or remain as silencers?

• The introduction nicely talks about enhancer grammar, and mentions the billboard and enhanceosome models. The authors should also include the Transcription Factor Collective model (Junion Cell 2012) and the dependency grammar model (Jindal and Farley Developmental Cell 2021)

• In their second library, the authors find pairwise interactions between CRX and NEUROD1, and CRX and RORB contributed to stronger enhancer activity. Do the authors find examples of these combinatorial regulatory logics within known photoreceptor enhancers? Or can they find co-occurrences of these motifs within putative regulatory elements as marked by ATAC-seq or other epigenetic marks?

Reviewer #2: The review comments are uploaded as an attachment.

Reviewer #3: This study builds on an intriguing observation from a previously published MPRA dataset: In rod photoreceptor cells, MPRA reporter activity has a nonmonotonic dependence on homotypic CRX binding sites (it increases from 0-2 binding sites and then decreases with 3-4 binding sites), while addition of NRL binding sites to sequences with 1-3 CRX sites enhances transcriptional activity relative to homotypic CRX binding (Fig. 1B). To investigate this further, Loell et al. fit MAVE-NN models to the MPRA data, choosing to investigate the MAVE-NN with pairwise TF interaction terms. Consistent with the data, the wild-type retina model shows negative pairwise interactions between strong homotypic binding sites (either NRL or CRX) and positive interactions between heterotypic binding sites (NFL and CRX). To explore this concept further, Loell et al. generate new data, constructing an MPRA library in which CRX binding sites are combined in sequences containing 1-2 other TF motifs (NRL, NeuroD1, RORB and MAZ). Again, a pairwise MAVE-NN model is fit to the data and the TF interaction terms are explored. Finally, the results are summarized in a very simple model that (1) reproduces the CRX- and NRL-dependent enhancer activity in Fig. 1B and (2) simulates enhancer behavior as a function of CRX sites and number of non-CRX or heterotypic binding sites, under the hypothesis that an increasing number of homotypic binding sites are repressive and that this repressive activity can be overcome by the binding of other TFs (heterotypic interactions).

The “underlying hypothesis” is clear: an increasing number of homotypic CRX binding sites are repressive and that this repressive activity can be overcome by the binding of other TFs (heterotypic interactions). However, the study results don’t provide sufficient support for the underlying hypothesis. At the end of the day, it is not clear that a generalizable regulatory grammar has been elucidated, limiting impact of the work.

Major concerns:

There are discrepancies between the MAVE-NN models fit to dataset 1 (NFL and CRX binding sites only) and dataset 2 (CRX, NFL, NeuroD1, RORB and MAZ).

• While the “dataset 1” model parameters support the “underlying hypothesis”, showing negative interaction terms for homotypic interactions between CRX and NFL and positive terms for heterotypic binding sites (Fig. 2D), the “dataset 2” model (Fig. 3D) loses the negative homotypic interaction for NFL, and a positive heterotypic interaction between 1 pair of TFs (CRX and RORB) binding sites is observed (interactions between CRX and the other three TFs are close to zero – no interaction, including for NFL, in contrast to the dataset 1 model – and the Crx-/- model in Fig. 2E). Thus, the simplified models, presented in Fig. 4A and 4B, are poorly supported by the “dataset 2” model.

• The disagreement between NFL-NFL pairwise interactions in the MAVE-NN model for dataset 1 versus 2 could be due to (1) differences in the MPRA reporter design and/or (2) overfitting of the data by the MAVE-NN models. Regardless, the generalizability of the underlying hypothesis based on the data and models in the study remains unclear. (If the NFL-NFL homotypic and heterotypic NFL-CRX interactions change so much between dataset 1 and dataset 2 models, how confident can we be in the other homotypic and heterotypic CRX interactions?) It would be instructive to visualize “dataset 2”, as was done for “dataset 1” in Fig. 1B, examining each TF separately, to see the synergistic impacts of the other TFs on enhancer activity of sequences containing the CRX sites. This will help determine whether the issue is (1) differences in reporter architecture and context or (2) a limitation of the MAVE-NN model.

• Thus, there is a need to find additional support for the simple models put forth in Fig. 4A and Fig 4B. There are several MPRA datasets and models that could examine the hypothesis further, specifically looking at CRX or CRX TF family members (comparing whether the number of homotypic binding sites shows a similar impact on enhancer activity and how heterotypic binding sites change this relationship). (See publications of Carl de Boer, data and existing models (e.g., in yeast, 2020 Nature), also CAGI challenge data and models). Of course, generalizability of the simple models to endogenous contexts would increase the impact of the work further, and that could be accomplished through motif analysis of nascent transcription data…

• It would be equally interesting to see where the underlying hypothesis holds and where it breaks down. My concern based on the manuscript in its current state is that the hypothesis was derived from published dataset 1 and that the analysis of the second dataset does not actually support the hypothesis…

If the MAVE-NN model of dataset 2 is robust and accurate, then the interactions describing homotypic and heterotypic interactions are more complex than presented in Fig. 4.

• From Fig. S3C, we see that the interaction term between high-affinity NRL and high-affinity CRX depends on the spacing between the binding sites, as there is a positive interaction when the binding sites are adjacent but not when there’s a greater spacing. Several of the other heterotypic interactions also show spacing-dependent interactions.

• The results for moderate affinity binding sites are often different from the high affinity RFX result, adding another layer of complexity. What is the biological interpretation?

• The simple model in Fig. 4 is at odds with the more complex picture from the pairwise MAVE-NN analysis.

Loell et al. need to demonstrate that the MAVE-NN models are robustly fit. There is no justification for the parameters selected for the MAVE-NN models nor are the results shown to be robust to the parameter choices.

The design of the dataset 2 MPRA is circular. NRL, NEUROD1, RORB and MAZ motifs were selected from sites identified in genomic strong enhancers that lose activity when the corresponding motif is deleted. The hypothesis from Fig. 1B is that binding of these other TFs can overcome repression of multiple CRX binding sites... but, given the selection criteria for the other TFs, this impact might be independent of CRX binding sites.

Other concerns:

• It would be beneficial to explain where the distribution of interactions from Fig. 3E came from – is it a distribution across positions?

• For comparison of MAVE-NN models (Fig. 2A, Fig. S2A), additional metrics of model prediction should be included in the comparison (e.g., Pearson correlation of prediction).

• For the “dataset 1” pairwise model, there’s a line of strong positive (or negative) interactions between moderate affinity reverse-strand CRX motif (m) and all other factors in adjacent positions (both Fig. S2D and E). What is the biological explanation for that or is there a problem with the way the model is explaining variance?

**Have the authors made all data and (if applicable) computational code underlying the findings in their manuscript fully available?**

Reviewer #1: Yes

Reviewer #2: None

Reviewer #3: Yes

PLOS authors have the option to publish the peer review history of their article (what does this mean?). If published, this will include your full peer review and any attached files.

Reviewer #1: No

Reviewer #2: No

Reviewer #3: No
---

## [Decision Letter · Decision Letter 1]

17 Sep 2023

Dear White,

Thank you very much for submitting your manuscript "Transcription factor interactions explain the context-dependent activity of CRX binding sites" for consideration at PLOS Computational Biology. As with all papers reviewed by the journal, your manuscript was reviewed by members of the editorial board and by several independent reviewers. The reviewers appreciated the attention to an important topic. Based on the reviews, we are likely to accept this manuscript for publication, providing that you modify the manuscript according to the review recommendations.

Sincerely,

Jie Liu

Academic Editor

PLOS Computational Biology

Jian Ma

Section Editor

PLOS Computational Biology

Reviewer's Responses to Questions

**Comments to the Authors:**

Reviewer #2: I am satisfied with the authors response to my comments and can recommend this for acceptance.

Reviewer #3: I commend the authors for thoughtful responses to the critiques, additional analyses and improved communication of the work. The manuscript is much strengthened by the revision.

Minor concerns:

• Fig. 3D – the x-axis labels are shifted relative to the bar plots.

• For the HepG2 and liver MPRA datasets, “pooled data” means that the RNA and DNA counts were summed as opposed to averaged across replicates? Please clarify. (Analysis of these additional contexts greatly strengthened the work.)

• Authors describe parameter uncertainty estimates in figure legends (e.g., Fig. S4C) as “standard parametric bootstrap replicates”. However, based on reading the methods (model fitting paragraph in Methods), “bootstrapping” (subsampling the underlying data with replacement to generate a distribution of parameters) was not performed. The methodology for modeling parameter uncertainty does not subsample the underlying MPRA data and therefore does not account for variability / noisy datapoints in that data. Parameter uncertainty needs to be achieved by subsampling / bootstrapping the underlying data and fitting multiple models and assessing parameter distributions across subsamples. Here, variability in parameters is estimated by fitting data simulated by a single MAVE-NN model fit to all data, so the parameter uncertainty is underestimated by this method, as (1) all data was used to fit the MAVE-NN model used for simulation (so no assessment of biological or technical variability of data) and (2) the MAVE-NN model is a simplified representation of the biology and therefore simulated datasets are unlikely to be representative of biological datasets. This impacts figures showing variability in parameters, where parameter uncertainties are likely underestimates. In terms of addressing this issue, I don't think the uncertainty estimates need to be re-evaluated, just clarify what was done and limitations. Authors could simply not using the word bootstrap (if bootstrap was not performed) in results / figure legends and instead pointing readers to methods. They might also add a sentence about caveats of their uncertainty measurements (underlying MPRA data was not subsampled, likely lowerbound on uncertainty), so readers are informed.

Thoughts for consideration:

• Would be nice to also visualize the Crx-/- MPRA data as was done for Fig. 1B. This will provide useful intuition that complements model parameters in Fig. 2. It would also be compelling to visualize the data this way for the other MPRA datasets modeled in this study, as these would support the “simple model” represented in Fig. 4.

• I like the idea of simple models from Fig. 4 being used to analyze, e.g., ATAC-seq data, predicting whether chromatin accessibility regions are likely to be enhancers or silencers based on counts of homotypic and heterotypic motif occurrences. Could be the focus of future work, but it would be nice to know qualitatively (perhaps using Spearman correlation) to quantify agreement of this simple model with the 5 MPRA datasets used in this study. If the simple models perform reasonably, this would be big for transcriptional regulatory network inference using ATAC-seq data.

Reviewer #4: The revised manuscript shows significant improvement and the authors have done a commendable job in addressing the previous comments. However, before its acceptance, I would like to suggest a few minor modifications to further enhance the clarity and completeness of the manuscript:

1. The authors should provide additional clarification regarding the specificity of their results. Are all the results cell-specific? If yes, the authors should indicate it in the very beginning of the results.

2. The preprocessing procedures and assumptions of the MAVE-NN model need to be explicitly defined in the manuscript. This will ensure that the methodology is transparent and reproducible.

3. The rationale behind using MAVE-NN, currently located in the discussion section, would be more appropriately placed in the introduction. This would provide the necessary context before presenting the results of the MAVE-NN.

4. Could the authors specify the sample size for each dataset used in the study including real ones? This information is crucial for understanding the robustness of the predicted results.

5. Lastly, it appears that the contribution of CRX binding sites to the activation or repression of transcription is inferred from the sign of parameters. Are these conditional activations or repressions? I recommend that the authors discuss the limitations of such interpretations in the discussion.

I believe addressing these minor issues will further strengthen the manuscript and make it ready for publication.

**Have the authors made all data and (if applicable) computational code underlying the findings in their manuscript fully available?**

Reviewer #2: Yes

Reviewer #3: Yes

Reviewer #4: Yes

PLOS authors have the option to publish the peer review history of their article (what does this mean?). If published, this will include your full peer review and any attached files.

Reviewer #2: No

Reviewer #3: No

Reviewer #4: No

Figure Files:

Data Requirements:

Reproducibility:

References:

---

## [Decision Letter · Decision Letter 2]

6 Jan 2024

Dear Dr. White,

We are pleased to inform you that your manuscript 'Transcription factor interactions explain the context-dependent activity of CRX binding sites' has been provisionally accepted for publication in PLOS Computational Biology.

Best regards,

Jie Liu

Academic Editor

PLOS Computational Biology

Jian Ma

Section Editor

PLOS Computational Biology

Reviewer's Responses to Questions

**Comments to the Authors:**

Reviewer #3: Thank you for the thoughtful revisions and responses. My concerns have been addressed.

Reviewer #4: I am content with the author's response; however, I was unable to locate information on the library size for each MPRA library at lines 702-706. It would be beneficial if the authors could create a table detailing the data sources. Besides this, I recommend acceptance.

**Have the authors made all data and (if applicable) computational code underlying the findings in their manuscript fully available?**

Reviewer #3: Yes

Reviewer #4: Yes

PLOS authors have the option to publish the peer review history of their article (what does this mean?). If published, this will include your full peer review and any attached files.

Reviewer #3: No

Reviewer #4: No

---

## [Editor Report · Acceptance letter]

11 Jan 2024

PCOMPBIOL-D-23-00357R2 

Transcription factor interactions explain the context-dependent activity of CRX binding sites

Dear Dr White,

I am pleased to inform you that your manuscript has been formally accepted for publication in PLOS Computational Biology. Your manuscript is now with our production department and you will be notified of the publication date in due course.

With kind regards,

Anita Estes
